# Cost-effectiveness analysis of an active 30-day surgical site infection surveillance at a tertiary hospital in Ghana: evidence from HAI-Ghana study

Evans Otieku [1,2] Ama Pokuaa Fenny,[1] Felix Ankomah Asante,[1] Antoinette Bediako-Bowan,[3,4] Ulrika Enemark[2]

[1]Economics Division, Institute of Statistical, Social and Economic Research, University of Ghana, Legon, Greater Accra, Ghana
[2]Department of Public Health, Aarhus University, Aarhus, Denmark
[3]Department of Surgery, Korle Bu Teaching Hospital, Accra, Ghana
[4]Department of Surgery, University of Ghana Medical School, Accra, Ghana

**Correspondence to**
Evans Otieku;
otieku@yahoo.com

## ABSTRACT

**Objective** To assess the cost-effectiveness of an active 30-day surgical site infection (SSI) surveillance mechanism at a referral teaching hospital in Ghana using data from healthcare-associated infection Ghana (HAI-Ghana) study.

**Design** Before and during intervention study using economic evaluation model to assess the cost-effectiveness of an active 30-day SSI surveillance at a teaching hospital. The intervention involves daily inspection of surgical wound area for 30-day postsurgery with quarterly feedback provided to surgeons. Discharged patients were followed up by phone call on postoperative days 3, 15 and 30 using a recommended surgical wound healing postdischarge questionnaire.

**Setting** Korle-Bu Teaching Hospital (KBTH), Ghana.

**Participants** All prospective patients who underwent surgical procedures at the general surgical unit of the KBTH.

**Main outcome measures** The primary outcome measures were the avoidable SSI morbidity risk and the associated costs from patient and provider perspectives. We also reported three indicators of SSI severity, that is, length of hospital stay (LOS), number of outpatient visits and laboratory tests. The analysis was performed in STATA V.14 and Microsoft Excel.

**Results** Before-intervention SSI risk was 13.9% (62/446) as opposed to during-intervention 8.4% (49/582), equivalent to a risk difference of 5.5% (95% CI 5.3 to 5.9). SSI mortality risk decreased by 33.3% during the intervention while SSI-attributable LOS decreased by 32.6%. Furthermore, the mean SSI-attributable patient direct and indirect medical cost declined by 12.1% during intervention while the hospital costs reduced by 19.1%. The intervention led to an estimated incremental cost-effectiveness ratio of US$4196 savings per SSI episode avoided. At a national scale, this could be equivalent to a US$60162248 cost advantage annually.

**Conclusion** The intervention is a simple, cost-effective, sustainable and adaptable strategy that may interest policymakers and health institutions interested in reducing SSI.

## Strengths and limitations of this study

► The paper used quality data from the healthcare-associated infection Ghana study and employed a widely accepted methodology and reporting standard for economic evaluation of healthcare interventions.
► To the best of our knowledge, the paper is the first to assess the cost-effectiveness of an active 30-day SSI surveillance in Ghana and add new evidence to the literature.
► Nonetheless, our measurement of effectiveness did not cover the entirety of health consequences attributable to SSI interventions. For example, we could not measure SSI-attributable quality-of-life years due to data limitations.

## INTRODUCTION

Hospital-acquired surgical site infection (SSI) is the most reported healthcare-associated infection (HAI) worldwide. The risk of SSI varies by region, country and healthcare facility. In developing countries, the risk averages 5.6 per 100 surgical procedures. Likewise, a study by the WHO also shows that up to one-third of all surgeries in low and middle-income countries (LMIC) lead to SSI.[1–4]

Besides the risk of infection, nosocomial SSI associates with increased morbidity, extended length of hospital stay (LOS) and costs to patients, society and health systems.[5 6] Published evidence suggests that the associated patient and hospital costs of SSI is about two times compared with patients without SSI.[7 8] A recent systemic review of the costs of SSI in 15 LMIC and 16 European countries finds that the additional attributable SSI cost is between US$174 and US$29610 in LMIC and US$21 and US$34000 in Europe.[9]

Potential SSI-attributable cost savings may result from an effective intervention strategy, which minimises the risk of infection and the consequent additional LOS, and extra cost of care was provided to affected patients. In addition, such interventions may positively

impact on patient health and well-being and loss of life. Consequently, it is of interest to identify interventions where the cost savings on treatment outweigh the cost of implementation, or where the cost of implementation net of savings is low compared with the health gains. For policymakers, the most effective but less costly intervention strategy is worth investing in.[1]

Several studies exist on the cost-effectiveness of various types of SSI interventions such as wound-edge protection devices after laparotomy,[10] negative blood pressure therapy,[11] bacterial binding dressing[12] and surveillance[8] with the latter showing the most promising results. The economic evaluation of SSI surveillance in England including feedback to surgical teams found a potential for significant cost savings accompanied by a decrease in SSI risk.[8] These economic evaluations are, however, from high-income country settings and the results may not be directly transferable to a lower middle income setting like Ghana.

The HAI-Ghana project is an international, multi-centre and interdisciplinary HAI project aiming to evaluate, among other things, the prevalence and costs of HAIs in Ghana and to identify and assess effects and costs of control interventions associated with selected HAIs (www.https://haiproject.org/index.php). As part of the HAI-Ghana project, a 30-day SSI surveillance with periodic feedback to surgical staff including SSI at the Korle Bu Teaching Hospital in Ghana was implemented in 2017–2018.[13] Before that, Ghana has had no surveillance strategy for monitoring SSI outbreaks. However, a 1-day point prevalence study in 2016 puts HAI prevalence in Ghana at 8.2%.[2] The choice of a 30-day surveillance follows the standard methods applied by the US Centres for Disease Control and Prevention National Health Care Safety Network and is used for SSI surveillance study in three African countries, that is, Kenya, Uganda and Zambia.[3]

Thus, the present study aimed to evaluate whether a 30-day SSI surveillance with periodic feedback to surgical staff could be a cost-effective strategy compared with a no surveillance practice. We excluded surgeries that had implants and will require longer than 30 days of follow-up.[13] We anticipated that increased awareness among hospital staff would reduce the risk of SSI and that the surveillance would also lead to early detection of SSI, both contributing to SSI-attributable cost savings. We further anticipated that cost of the intervention could be less or only slightly more than the savings, thus, making a surveillance system a worthwhile investment from a societal point of view.

## METHODS
### Design
We conducted a cost-effectiveness analysis of an SSI surveillance intervention using a simple decision model with parameters based on a before and during-intervention study design. The data collection before intervention took place from June to September 2017. The intervention was implemented from October 2017 to December 2018. Cost data collection during intervention took place between April and July 2018. The lag time of 6 months between the two studies was based on expert medical consultation of the need for medical staff to fully acquaint themselves with the surveillance strategy. The effect was evaluated based on follow-up until December 2018. By design, we could only measure results during the intervention and not after because the active surveillance requires selection of participants while the intervention was ongoing.

For methodological transparency, we report according to the Consolidated Health Economic Evaluation Reporting Standard checklist.[14]

### Study setting
The intervention took place at the general surgical unit (GSU) of the Korle-Bu Teaching Hospital (KBTH) in the Greater Accra region of Ghana. The GSU is a 150-bed capacity equivalent to 7.5% of the total bed capacity of the KBTH. An estimated average of 2280 surgical operations is performed at the GSU annually, equivalent to 0.9% of surgeries performed yearly in Ghana.[15] At the time of the study, the surgical department (SD) of the hospital had 12 active general surgeons, 8 seniors and 16 junior residents under surgical training. There were also 12 interns and o158 nurses and healthcare assistants.[13]

### Target population and subgroup
The study population includes all patients who underwent surgery at the GSU of the KBTH. Exclusion criteria were patients who underwent surgical procedures where primary closure of incision was not completed in the theatre of the SD or had an implant surgery.[5 13] A total of 446 and 582 patients were eligible for inclusion before and during the intervention studies, respectively. Among these, 62 and 49 patients with SSI within 30 days after surgery based on 2017 SSI criteria were defined by the Centers for Disease Control and Prevention[16] and identified by medical staff through a healthcare personnel-based survey and a patient-based telephone survey.[13] We included SSI from superficial, deep and organ space wounds, resulting from 11 surgical procedures related to thyroid, breast, colon, appendix, limb amputation, etc. However, only 40 and 31 patients with SSI, respectively, are considered in the cost analysis because some declined to participate, a few absconded before and after discharge, and others were untraceable during the follow-up due to wrong contact information.[5]

### Patient and public involvement
Patients and public involvement in the study were twofold. First, a pretest of the validated data collection tool sought patients' opinions on the dimensions of the direct and indirect cost associated with SSI. The aim was to ensure that the study design and data capture reflect patients' and carers' preferences and priority spending

on SSI-related costs. Patients' involvement in the recruitment and conduct of the study also include their time to provide data during the 30-day follow-up. Second, the management of the HAI-Ghana Project organised two separate seminars/conferences postbaseline study (before intervention) to educate patients, hospital staff, the scientific community and the general public on SSI-associated costs. The conference provided an opportunity for diverse suggestions on the appropriate SSI intervention to implement within the LMIC setting. It also raised public awareness and motivation for investment in SSI interventions. The before-intervention study is published elsewhere[5] as part of the project dissemination plan.

## Study perspective

The baseline study evaluated the attributable costs of SSI from the patient and provider perspectives. Therefore, this study measures the cost-effectiveness of the intervention from the same perspectives. The provider refers to the KBTH in Ghana, where the study took place.

## Description of the intervention (comparators)

Before June 2017, there was no SSI surveillance strategy at the SD of the hospital. Both inpatient and outpatients who underwent surgical procedures at the GSU received standard care postsurgery, which is a periodic review of health state and wound dressing by medical staff until the surgical wound was healed or poses no threat to their health. In the before-intervention study, patients who received standard care at the SD were prospectively enrolled after surgery up to 30 days.

An intervention involving an active 30-day SSI surveillance strategy was rolled out from October 2017 to December 2018. Prior to the start, medical staff at the SD were informed about the alarming rate of SSI. Interns at the SD were trained to continuously monitor all surgery patients. For inpatients, the surveillance was done by daily inspection of the surgical wound area for 30-day postsurgery. The inpatient surveillance was discontinued when a patient was transferred out of the GSU postsurgery, died or the 30-day surveillance ended while the patient was still in the hospital. For postdischarge patients, the surveillance involved both healthcare personnel-based surveillance and patient-based telephone surveillance. In the case of the healthcare personnel-based surveillance, each patient, at discharge, was given a wound card used by nurses or healthcare assistants to document the state of the wound. Each surgical ward has a wound treatment room where patients receive wound dressing postdischarge. For the patient-based telephone survey, discharged patients were followed up by phone call on postoperative days 3, 15 and 30 using a recommended surgical wound healing postdischarge questionnaire designed by the Surgical Site Infection Surveillance Service of Public Health England.[17] The questionnaire consists of a series of 'yes' or 'no' questions to help diagnose SSI for patients who noticed any changes in their wounds. If SSI was suggested, diagnosis criteria were documented and confirmed at a change of

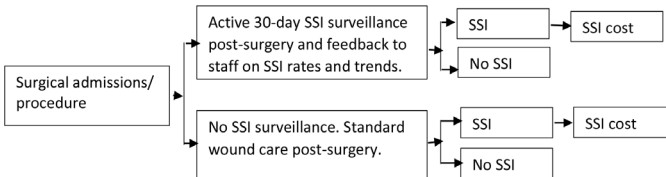

**Figure 1** Decision tree. SSI, surgical site infection.

wound dressing. Thus, compliance with the intervention followed an infection prevention and control protocol and was monitored by a project team member (BA) who is also a trained general surgeon and a staff of the University of Ghana Medical School and KBTH. Staff (surgeons only) received quarterly feedback on SSI rates at the SD units, across risk factors and over time.

## Decision model

Our simple decision tree model compares a cohort of surgical patients with and without exposure to the surveillance intervention (figure 1). The assumption underpinning the model is that the success of the active 30-day SSI surveillance will lead to: (1) a reduction in the probability of SSI, which will reflect in a reduction in the number of SSI episodes and the associated treatment costs and (2) possibly early detection of SSI, which could be reflected in lower treatment cost per case because of shorter length of stay. The parameter values used in the model are based on the observed probability of SSI before and during intervention, the estimated SSI-attributable costs and the intervention costs (table 1). For comparison of costs before and during the intervention, a discount of 2.5% was taken into account.[18]

## Measurement of effectiveness

Our primary effect measure is the number of SSI cases avoided with the intervention (avoidable SSI morbidity risk) and the associated costs. We believed the number of deaths avoided and life years lost based on the estimated change in mortality risk is considerably more uncertain given the low number of observations. Thus, measurement of effectiveness took into account the incremental outcome associated with the intervention and estimated as:

$$\text{Effectiveness} =$$

$$\left[ \text{During} - \text{intervention SSI}_{\text{outcome}} - \text{Before} - \text{intervention SSI}_{\text{outcome}} \right]$$

$$(1)$$

## Estimating resources and costs

To determine the SSI-attributable costs, we matched SSI—patients with non-SSI patients based on age, sex, surgical procedure, etc. See Fenny *et al* for detail on matching criteria.[5] For precision, we used an activity-based microcosting-dubbed ingredient costing method to capture and measure patient costs.[5] Patient cost data were collected using a validated questionnaire that covered medical and non-medical costs incurred by patients with SSI and their caregivers for the period of hospitalisation and up to 30 days after surgery. It includes total direct

**Table 1** Model parameter values

| Parameters | Value (95% CI) | Source |
|---|---|---|
| Probability of SSI before intervention (%) | 13.9 | Fenny et al[5] |
| Probability of SSI during intervention (%) | 8.4 | Patient data* |
| Patient cost of SSI before intervention (US$) | 2208 (2087 to 2376) | Fenny et al[5] |
| Patient cost of SSI during intervention (US$) | 1942 (1758 to 2227) | Patient data* |
| Hospital cost before intervention (US$) | 1391 | Fenny et al[5] |
| Hospital cost during intervention (US$) | 1047 | Patient data* |
| Cost discount rate (%) | 2.5 | GSS[17] |

*Estimate from patient data during-intervention study.
GSS, Ghana Statistical Service; SSI, surgical site infection.

and indirect medical expenses. The direct medical cost comprises expenses related to laboratory tests, medical consultation, review/outpatient care cost and drugs. The indirect medical costs were those related to transportation, feeding and accommodation. Public health facilities in Ghana run a cost-recovery system, in which the cost of direct medical services like medicines, consultation and laboratory tests is paid either out-of-pocket or by reimbursement through the national health insurance scheme. Furthermore, we included in our analysis the indirect cost of SSI due to productivity loss, resulting from absenteeism from work. Thus, the indirect cost of SSI captures the loss of income during inpatient days and up to 30 days after surgery or death.[5]

For the measurement of the hospital cost, the estimated daily hospital cost of US$325 and US$390 from the before and during intervention per patient was multiplied by the mean extra SSI-attributable LOS of 4.6 and 3.1 days, respectively. The procedure used to estimate the hospital cost is published in the before-intervention study on the cost of SSI.[5] Briefly, the procedure involves an activity-based gross costing that captures the sum of all recurrent and annualised capital expenditures incurred by the GSU within the 2017/2018 financial year. The recurrent cost includes staff remuneration, cost of clinical support and all other consumable items used by the GSU. The capital cost comprises the annualised expenditures of office space, including the theatre room, patient wards, changing rooms, etc.

The intervention cost was estimated as the sum of all expenses incurred during the intervention, and which would be necessary for continuation of the intervention. It includes the cost of wound cards, wound swaps, laboratory analysis of wound swaps, airtime for postdischarge surveillance and the value of extra time spent on SSI surveillance. Materials were valued at market price and the extra time spent was valued using a per diem amount. Costs specific to the research project, for example, additional data collection, were excluded. The cost of the intervention was absolved by the HAI-Ghana project.

## Data analysis

First, to assess the comparability of the before-intervention and during-intervention groups, we evaluated statistical differences in background characteristics of the participants using non-parametric $\chi^2$ statistic for categorical variables (sex, wound class and surgical procedure) at an alpha of $p<0.05$. Second, we estimated SSI cases avoided as the difference in SSI risk ratio before and during the intervention and report three indicators of severity, that is, mean LOS, number of laboratory tests and outpatient visits. Third, we present information on the total cost of the intervention and estimated the SSI-attributable patient and hospital/provider cost per SSI case. The SSI-attributable costs before intervention was obtained from our previous paper.[5] Cost were calculated in local currency (Ghana cedi) and converted into base year (2018) Purchasing Power Parity (PPP) in US Dollars (US$) using a web-based PPP convertor that equates US$1.00 to Ghana cedi (GHC) 1.569.[19] We applied a 2.5% discount rate[18] to adjust the 2017 baseline costs to 2018 levels.

Third, we calculated the incremental cost savings from the intervention across domains: the mean direct medical and non-medical cost and indirect cost (productivity loss). Finally, we populated our decision model with the estimated parameter values using the during-intervention cohort as basis for calculating costs and effects with and without the surveillance intervention. We estimated the incremental cost-effectiveness ratio (ICER) by dividing the change in total cohort cost by the effects gained, that is, the number of SSI cases avoided (SSI morbidity risk avoided). The estimation of ICER assumed the formula:

$$ICER = (C_1 - C_0) / (E_1 - E_0) \qquad (2)$$

Thus, $C_1$ and $C_0$ are the total cohort cost of the intervention and the comparator, while $E_1$ and $E_0$ represent the effect gain from the intervention and the comparator, respectively.[20 21] The whole analysis was performed in STATA V.14.0 and Microsoft Excel. We deployed inbuilt statistical analysis tools in STATA V.14.0 to compute mean values and 95% uncertainty intervals (95% UI).

## Sensitivity analysis

The robustness of the incremental cost savings derived from the intervention was evaluated using both one-way and multiway sensitivity analysis. Cost input parameters were varied using the minimum/maximum values of the 95% UI around the base case mean cost. The input parameters include patient direct cost, cost due to productivity loss, hospital cost and the probability of SSI before and during the intervention. The assumption is that a significant change in the input parameters may affect the incremental cost advantage and consequently nuance possible conclusions and recommendations of the study.[22] Microsoft Excel allowed us to build a sensitivity analysis table and vary the input parameters to determine a change in incremental cost savings derived from the intervention.

## RESULTS

Enrolled patients in the before and during-intervention studies were 446 and 582, respectively. Of the patients identified with SSI, we included 40 and 31 in the cost analysis because some declined to participate, while others were lost to follow-up. In both data collection periods, women were more than half of the eligible participants.

Approximately, 45% and 32% of patients with SSI in the before and during-intervention groups had dirty surgical wounds. Eleven surgical procedures were examined. Overall, limb amputation and breast surgery dominated in both the control group (before intervention) and the case group. Among patients with SSI, those who underwent herniorrhaphy surgery were 16.1% in the case group and 19.7% in the control group (table 2).

## Incremental outcomes

SSI morbidity risk was 13.9% (62/446) in the before-intervention phase but declined to 8.4% during-intervention phase (49/582), equivalent to a risk difference of 5.5% (95% CI 5.3 to 5.9). The mortality risk of SSI also declined by 33.3%, while the mean additional LOS attributable to SSI declined by 32.6% during intervention. Thus, the latter suggests that early detection of SSI may have reduced treatment needs, which may be reflected in reduced LOS. Furthermore, the mean number of outpatient visits for surgical wound care reduced from 7.9 (95% CI 7.3 to 8.5) to 5.2 (95% CI 4.4 to 5.6) in 30-day postsurgery. Likewise, the mean number of laboratory test for patients with SSI decreased by 48% (table 3).

| Table 2 Descriptive characteristics of respondents | | | | | |
|---|---|---|---|---|---|
| | **Before intervention** | | **During intervention** | | **P value comparing SSIs for before and during intervention** |
| | **Total (n=446)** | **SSI (n=40)** | **Total (n=582)** | **SSI (n=31)** | |
| Mean age in years(95% CI) | 46 (45-48) | 41 (40-43) | 43 (41-44) | 44 (42 – 47) | |
| Sex of neonates | | | | | |
| Male | 171 (38.0) | 9 (22.5) | 256 (44.0) | 12 (38.8) | p=0.083 |
| Female | 275 (62.0) | 31 (77.5) | 326 (56.0) | 19 (61.2) | |
| Wound class | | | | | |
| Clean | 289 (64.7) | 10 (25.0) | 319 (54.8) | 7 (22.6) | |
| Clean contaminated | 30 (6.7) | 3 (7.5) | 66 (11.3) | 2 (6.5) | p=0.172 |
| Contaminated | 57 (12.7) | 9 (22.5) | 144 (24.7) | 12 (38.7) | |
| Dirty | 70 (15.9) | 18 (45.0) | 53 (9.1) | 10 (32.2) | |
| Procedure | | | | | |
| Thyroid surgery | 40 (9.0) | 3 (7.5) | 40 (6.9) | 2 (6.5) | |
| Breast surgery | 64 (14.3) | 7 (17.5) | 187 (32.1) | 7 (22.6) | |
| Herniorrhaphy | 88 (19.7) | 11 (27.5) | 116 (19.9) | 5 (16.1) | |
| Gall bladder surgery | 5 (1.1) | 2 (5.0) | 12 (2.1) | – | |
| Bile duct/liver surgery | 3 (0.7) | – | 7 (1.2) | – | |
| Small bowel surgery | 8 (1.8) | 2 (5.0) | 5 (0.9) | 1 (3.1) | p=0.295 |
| Appendix surgery | 69 (15.5) | 6 (15.0) | 86 (14.8) | 5 (16.1) | |
| Colon surgery | 16 (3.6) | 5 (12.5) | 9 (1.5) | 3 (9.7) | |
| Rectal surgery | 8 (1.8) | 1 (2.5) | 7 (1.2) | 2 (6.5) | |
| Laparotomy* | 47 (10.5) | 1 (2.5) | 101 (17.3) | 3 (9.7) | |
| Limb amputation | 98 (22.0) | 2 (5.0) | 12 (2.1) | 3 (9.7) | |

*Laparotomy other than above abdominal surgeries.
SSI, surgical site infection.

**Table 3** Incremental outcomes before and during intervention for patients with SSI

| Sample description and outcome | Before intervention (95% CI) | During intervention (95% CI) | Incremental (%) | P value* |
|---|---|---|---|---|
| Number of SSI cases identified | 62 (13.9) | 49 (8.4) | – | |
| SSI mortality risk (%) | 4.8 | 4.1 | −1 (33.3) | 0.077 |
| Indicators of severity | | | | |
| Mean LOS for the overall sample | 8.3 | 7.1 | −1.2 (13.0) | 0.091 |
| Mean additional LOS | 4.6 (3.8 to 5.3) | 3.1 (2.9 to 3.4) | −1.5 (32.6) | <0.05 |
| Mean number of outpatient visits | 7.9 (7.3 to 8.5) | 5.2 (4.4 to 5.6) | −2.7 (34.2) | <0.05 |
| Mean number of laboratory test | 2.5 (2.2 to 2.9) | 1.3 (1.1 to 1.6) | −1.2 (48.0) | <0.01 |

*Comparing before and during-intervention outcomes.
LOS, length of hospital stay; SSI, surgical site infection.

## Intervention cost

Table 4 depicts the activity-based intervention cost in terms of hospital resources used on implementation of the surveillance system. Approximately, 46% of the intervention cost was related to per diem expenses for the extra time spent by interns and staff on SSI surveillance for the duration of the study. We derived the per-patient cost of the intervention by dividing the grand total cost of the intervention US$10 750 by the overall sample of 582. Thus, the hospital cost per-patient included in the surveillance intervention amounts to US$18.47.

## SSI-attributable costs

We present the disaggregated SSI-attributable costs before and during intervention in eight domains. The mean expenses on antibiotics increased by 8.9% during intervention. Overall, the intervention led to a 12.1% decrease in the mean patient cohort cost per SSI episode. Furthermore, the hospital cost associated with SSI was reduced by 24.7%, equivalent to US$344.00 with the intervention (table 5).

## Cost-effectiveness

The sum of the mean patient cost and hospital cost per SSI episode multiplied by the number of patients with SSI plus the intervention cost for the cohort equals the total cost of care attributable to SSI in the intervention and control groups. The results show that the intervention is both cost saving and has positive effects and is, therefore, a dominant strategy (table 6). The estimated ICER amounts to US$4196 cost savings per SSI episode avoided.

## Sensitivity analysis

Figures 2 and 3 present the result for the uncertainty analysis of incremental cost savings from the intervention. The sensitivity analysis shows that for each cost input parameter, varying the values within the 95% UI may still lead to some cost savings with the intervention. The minimum and maximum cost savings of US$1910 and US$5720 may result if the probability of SSI before and during the intervention assumed the lower values of the 95% UI of 0.099 and 0.061 compared with the base values of 0.139 and 0.084, respectively.

## DISCUSSION

This paper evaluates the potential cost savings of implementing SSI intervention within resource-limited settings of LMIC with Ghana as a case study. We hypothesised that an active 30-day SSI surveillance strategy could primarily reduce SSI morbidity risk and consequently the associated mortality, LOS and costs, and that these savings would exceed the cost of the intervention.

The measurement of incremental outcomes shows probable health and economic benefits with the intervention. For instance, we found that the intervention resulted in approximately 14% and 33% reduction in SSI morbidity and mortality risks, which may mean additional benefit to patients if we consider the quality-of-life years gain from the intervention. Likewise, the decline in SSI-attributable LOS by almost 33% and its associated patient

**Table 4** Intervention cost (2018 PPP-adjusted US$)

| Description | Quantity | Unit cost | Total |
|---|---|---|---|
| Wound cards | 10 sets | 55 | 550 |
| Wound swab | 500 pieces | 1.28 | 640 |
| Laboratory analysis of wound swab | Four instalments | 960/instalment | 3480 |
| Airtime for post-discharge surveillance | Six set | 160 | 960 |
| Per diem for SSI surveillance* | 4 months | 1280/month | 5120 |
| **Grand total** | | | 10 750 |
| **Cost per patient of the intervention** | | | 18.47 |

*Comprises four interns and four nurses (matron-in-charge on each of the four surgical floors).
PPP, Purchasing Power Parity; SSI, surgical site infection.

**Table 5** SSI-attributable costs before and during intervention (2018 PPP-adjusted US$)

| | Before intervention US$ (95% CI) | During intervention US$ (95% CI) | Difference US$ (% change) | P values* |
|---|---|---|---|---|
| Patient direct medical costs | | | | |
| Mean cost of systemic antibiotics | 419.66 (398 to 458) | 456.82 (408 to 464) | 37.16 (8.9) | 0.219 |
| Mean cost of wound dressing | 74.82 (68 to 82) | 58.58 (46 to 62) | −16.24 (21.7) | 0.332 |
| Mean cost of laboratory tests | 117.16 (111 to 125) | 51.93 (46 to 54) | −65.23 (55.7) | <0.05 |
| Mean cost of consultation | 634.38 (593 to 677) | 574.24 (490 to 589) | −60.14 (9.5) | <0.05 |
| Patient indirect medical cost† | | | | |
| Mean non-medical cost | 380.63 (355 to 398) | 291.64 (260 to 316) | −88.99 (23.4) | <0.01 |
| Patient indirect costs | | | | |
| Mean cost of productivity loss | 581.60 (355 to 831) | 509.02 (309 to 660) | −72.60 (12.5) | 0.110 |
| Mean patient cost per patient with SSI | 2208.25 (2087 to 2376) | 1942.23 (1758 to 2227) | −266.02 (12.1) | |
| Hospital cost attributable to SSI | 1391 | 1047 | −344 (24.7) | |
| Total | 3599.25 | 2989.23 | −610 (16.9) | |

*P values comparing significant differences in SSI-attributable cost before and during intervention.
†Includes cost of transportation, accommodation and feeding.
PPP, Purchasing Power Parity; SSI, surgical site infection.

and provider costs by roughly 12% and 19% mean that the intervention is not only life-saving but cost saving and a dominant strategy. Thus, not only did the intervention reduce the incidence of SSI but patients with SSI also incurred relatively low average cost of treatment compared with a no surveillance scenario.

A recent study estimated that approximately 869 surgeries per 100 000 of the population are performed in Ghana[15] annually, resulting in 260 700 surgeries in 2020. Now, assume that similar surveillance interventions could be implemented with similar effects as in the present study. Applying the SSI morbidity risk of 13.9% and 8.4% as observed before and during intervention to the estimated total surgeries in 2020 suggests that

this may lead to 14 338 avoided SSI episodes, which may yield US$60 162 248 cost savings from both patient and provider perspectives. Notably, the intervention may free 98 803 extra patient bed-days attributable to SSI annually across the country, equivalent to about 13 916 new admissions. In settings like Ghana, where some prospective surgical patients could be put on a waiting list due to limited hospital-bed capacity,[23] the intervention may provide an opportunity to admit more surgical patients. Comparatively, our finding is congruent to other studies that have evaluated either the cost-benefit or cost-effectiveness of SSI surveillance in settings like England, the USA, Australia, among others.[8 24–26] In one such study involving more than 5.8 million participants selected

**Table 6** Incremental cost-effectiveness

| | Admitted to ward without surveillance (before-intervention figures applied) | Admitted to ward with surveillance (during-intervention figures applied) |
|---|---|---|
| Parameter values | | |
| Surgical patient cohort | 582 | 582 |
| Intervention costs | 0 | 10 750 |
| Probability of SSI | 0.139 | 0.084 |
| Mean SSI-attributable patient costs | 2208 | 1942 |
| Mean additional hospital costs | 1391 | 1047 |
| Results | | |
| Expected SSI-related costs | | |
| Patient | 178 643 | 94 952 |
| Hospital | 112 529 | 61 936 |
| Total | 291 172 | 156 888 |
| Expected number of SSI cases | 81 | 49 |

SSI, surgical site infection.

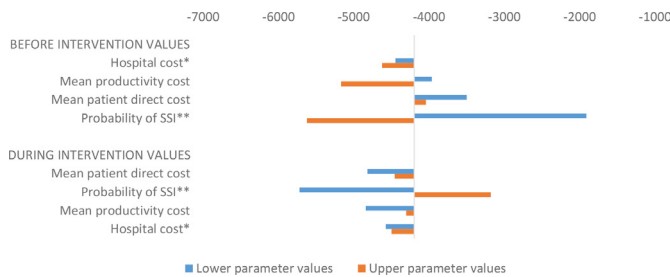

**Figure 2** Incremental cost savings using upper and lower parameter values (base case: –US$4,196). *Lower and upper values took into account cost adjustment rate of 2.5%.[17] **Values based on estimate from another study.[13] SSI, surgical site infection.

from Europe, Australia and Asia, the authors found that hospital-based SSI surveillance results in a 35% risk reduction in 9 years.[26]

In descending order of magnitude, the uncertainty analysis shows that the estimated patient cost savings attributable to SSI may be sensitive to the probability of SSI, cost of productivity loss, patient direct cost, etc. However, as expected, the intervention remains a dominant strategy irrespective of the uncertainties in cost input parameters. We argue that comparing the unit cost of implementing the intervention to its benefits makes a strong case for policymakers to absorb or subsidise the intervention cost. In places like Ghana, where a considerable share of the patient direct cost is covered by health insurance, it could be in the interest of the health insurance scheme to support such interventions. Alternatively, hospitals with the capacity to adapt and institutionalise the intervention may do so to reduce cost. Likewise, governments may be interested in supporting such interventions, which will free resources to admit more patients, thereby showing their capacity to improve service provision to their constituents.

Finally, our inability to include all eligible patients with SSI in the study was due to a situation beyond control and is ethically grounded since some declined to participate and others were lost to follow-up. As previously argued in the baseline paper on the cost of SSI,[5] the study took place in one teaching hospital in Ghana, which may not provide a general reflection of what pertains in other

settings. We observed significant differences in wound class and surgery type between groups of participants before and during the intervention, which may lead to an underestimation of the impact of the intervention. Also, the use of mean LOS and cost may not reflect the true measure of the estimated additional LOS and cost due to the intervention.[27] Even though compliance with the intervention was supervised and hypothesised to reduce the incidence of SSI at the hospital, we are unable to say whether the reduction in SSI incidence was occasioned mainly by the active 30-day surveillance. Therefore, further studies may help conclude on its effectiveness and provide current evidence to improve the lack of studies on economic benefits of SSI interventions in LIMC.[28]

## CONCLUSION

SSI is preventable if an effective intervention is in place. This paper presents an SSI intervention that is adaptable, cost-effective and sustainable within resource-limited settings. It highlights the potential gains and trade-offs that may inform decision-makers to consider an upscale of the intervention. Specifically, it shows that an active 30-days SSI surveillance may reduce the risk of SSI morbidity and mortality by approximately one-third and consequently reduce the patient cost by almost 12% and hospital cost by almost 25%.

**Acknowledgements** The authors acknowledge the support of the data collection team and the hospital staff of the General Surgical Department of the Korle-Bu Teaching Hospital, who assisted in the SSI surveillance. We also acknowledge the support of study participants (patients), the general public, and the scientific community for their contribution to the design, recruitment, conduct of the study, and dissemination of the findings.

**Contributors** EO, APF, UE and FAA conceived and designed the study. APF, FAA and UE reviewed the data collection tools. EO, APF and AB-B collected the data. EO and APF analysed the data. EO and APF drafted the manuscript. UE, FAA and AB-B critically reviewed and revised the manuscript. All authors read and approved the final manuscript. EO is the data curator and guarantor of the overall content of the manuscript.

**Funding** Financial support was provided by the Danish Ministry of Foreign Affairs as part of the HAI-Ghana project/DANIDA grant number 16-PO1-GHA.

**Competing interests** None declared.

**Patient and public involvement** Patients and/or the public were involved in the design, or conduct, or reporting, or dissemination plans of this research. Refer to the Methods section for further details.

**Patient consent for publication** Not applicable.

**Ethics approval** Ethics approval for this study was granted by the Korle-Bu Teaching Hospital Ethics Committee with reference number KBTH-IRB/0036/2017. All participants signed the informed consent form.

**Provenance and peer review** Not commissioned; externally peer reviewed.

**Data availability statement** Data are available upon reasonable request. Data used for this study is not publicly available but can be assessed upon reasonable request to the corresponding author due to ethics approval guidelines.

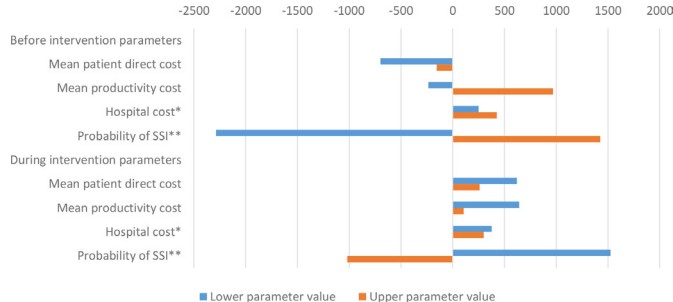

**Figure 3** Deviations in savings from base case: US$4196. *Lower and upper values took into account cost adjustment rate of 2.5%.[17] **Values based on estimate from another study.[13] SSI, surgical site infection.

ORCID iD
Evans Otieku http://orcid.org/0000-0002-6809-5160

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
