## [Reviewer comments · BMJ Open]

ARTICLE DETAILS

TITLE (PROVISIONAL)	Cost-effectiveness Analysis of an active 30-day Surgical Site Infection Surveillance at a Tertiary Hospital in Ghana: Evidence from HAI-Ghana Study
AUTHORS	Otieku, Evans; Fenny, Ama; Asante, Felix; Bediako-Bowan, Antoinette; Enemark, Ulrika

VERSION 1 – REVIEW

REVIEWER	Mao, Wenhui Duke Global Health Institute
REVIEW RETURNED	12-Oct-2021

GENERAL COMMENTS	This article assessed the cost-effectiveness of 30-days SSI surveillance in a teaching hospital in Ghana. This article brought important cost and effectiveness information, and hopeful inform the policy making process. However, the causal-relationship between intervention and observed effect is limited by the study design. Few comments below: please spell out the abbreviations in abstract (i.e.HAI) please describe the intervention in abstract please clarify which cost is measured, direct medical cost? Introduction: this session is informative and well organized. please consider add more background information of the SSI in Ghana. Methods: could you please justify, why a before and during intervention strategy was used? why not measure results after intervention? Could you please explain a bit more of the patient cost? is it out-of-pocket or total medical cost? what is the role of NHIS? Results: could you please present P value for Table 3 and 5? please clarify, only direct medical cost was measured in this study. but across the text, "societal" was used multiple times. Did you measure indirect cost as well?
---

REVIEWER	McFarland, Agi Glasgow Caledonian University, Nursing and Community Health
REVIEW RETURNED	01-Nov-2021

GENERAL COMMENTS	Many thanks for the opportunity to review your study. I have a few questions and some additions which would improve the current manuscript, below:  • No information is provided on ethical approval; please add this in • What diagnostic criteria did you use for SSI? And which type of SSI were included from which surgeries? • Why was the follow up limited to 30 days given that SSI may be diagnosed beyond that given current diagnostic criteria from CDC? • What software did you use for the modelling? Please specify • You mention discounting at 2.5% on page 5 but then refer to a 2.5% inflation using the same reference on page 6; it is not clear which you did and needs attention • Model parameter values need further detail in relation to units of measurement • There were significant differences in wound class breakdown and surgery types between your comparator groups which may have under estimated the overall impact of your intervention. This is worth highlighting in the discussion • Please also consider the limitations of using mean LOS comparisons; a useful article in this regard: https://pubmed.ncbi.nlm.nih.gov/29902486/ • Your point about the LMIC setting is an important one given the lack of such studies identified in this recent review: https://pubmed.ncbi.nlm.nih.gov/32417433/
---

VERSION 1 – AUTHOR RESPONSE

Reviewer 1

Comment 1: This article assessed the cost-effectiveness of 30-days SSI surveillance in a teaching hospital in Ghana. This article brought important cost and effectiveness information, and hopeful inform the policy making process. However, the causal-relationship between intervention and observed effect is limited by the study design.

Response: Thank you for this comment. We acknowledged the limitations of the design under subsection “Strength and limitations of the study”. (see page 2 main document).

Comment 2: Please spell out the abbreviations in abstract (i.e. HAI).

Response: All the abbreviations in the abstract are spelled out in the first instance and used subsequently in order not to exceed the word limit of the abstract.

Comment 3: Please describe the intervention in the abstract.

Response: Resolved

Comment 4: Please clarify which cost is measured, direct medical cost?

Response: Resolved. Kindly refer to page 9 and Table 5 of the main document file.

Comment 5: The introduction: This session is informative and well organized. please consider adding more background information on the SSI in Ghana.

Response: Resolved. See pages 3&4.

Comment 6: Methods: Could you please justify, why a before and during intervention strategy was used? why not measure results after intervention?

Response: Resolved under "Design".

Comment 7: Could you please explain a bit more of the patient cost? is it out-of-pocket or total medical cost? what is the role of NHIS?

Response: Resolved. Kindly see the section on “Estimating resources and costs”(pages 5 & 6).

Comment 8: Results: Could you please present P-value for Tables 3 and 5?

Response: Resolved.

Comment 9: Please clarify, only direct medical cost was measured in this study. but across the text, "societal" was used multiple times. Did you measure indirect cost as well?

Response: We measured both direct and indirect medical costs as patient costs and also report patient productivity loss due to absenteeism from work. (Cost breakdown is detailed in Table 5).

Reviewer 2

Comment 1: No information is provided on ethical approval; please add this in

Response: Ethics approval information is provided under subsection "Ethics Approval Statement."

Comment 2: a) What diagnostic criteria did you use for SSI?

b) And which type of SSI were included from which surgeries?

Response: a) SSI diagnosis was based on CDC criteria.

b) Types of SSI included were those related to superficial, deep, and organ space wounds resulting from eleven surgical procedures (Table 2).

Comment 3: Why was the follow-up limited to 30 days given that SSI may be diagnosed beyond that given current diagnostic criteria from CDC?.

Response: The 30-day surveillance follows standard methods described by the CDC. We excluded SSI from implant surgery, which may require more than 30-days follow-up.

Comment 4: What software did you use for the modeling? Please specify.

Response: We used both STATA version 14.0 and Microsoft Excel to perform the whole analysis.

Comment 5: You mention discounting at 2.5% on page 5 but then refer to a 2.5% inflation using the same reference on page 6; it is not clear which you did and needs attention.

Response: Resolved. This was an oversight. We used a 2.5% discount rate to compare costs before and during the intervention.

Comment 6: Model parameter values need further detail in relation to units of measurement.

Response: Resolved, thank you.

Comment 7: There were significant differences in wound class breakdown and surgery types between your comparator groups which may have underestimated the overall impact of your intervention. This is worth highlighting in the discussion.

Response: Resolved. See page 11 of the manuscript.

Comment 8: Please also consider the limitations of using mean LOS comparisons; a useful article in this regard: <https://pubmed.ncbi.nlm.nih.gov/29902486/>

Response: Resolved. See page 11

Comment 9: Your point about the LMIC setting is an important one given the lack of such studies identified in this recent review: <https://pubmed.ncbi.nlm.nih.gov/32417433/>

Response: Resolved. See page 11.

VERSION 2 – REVIEW

REVIEWER	McFarland, Agi Glasgow Caledonian University, Nursing and Community Health
REVIEW RETURNED	07-Dec-2021

GENERAL COMMENTS	Please specify the year of CDC diagnostic criteria and provide a reference. Study perspective: please be specific in who the "provider" is
---

	Please add in details of the statistical software used for the modelling into the Methods
--	---

VERSION 2 – AUTHOR RESPONSE

Reviewer 2

Comment 1: Please specify the year of CDC diagnostic criteria and provide a reference.

Response: Resolved (see pages 3).

Comment 2: Study perspective: please be specific in who the "provider" is

Response: Resolved (see page 4).

Comment 3: Please add in details of the statistical software used for the modelling into the Methods.

Response: Resolved (see pages 1 and 7)